# Effects of Dry-Hopping on Beer Chemistry and Sensory Properties—A Review

**DOI:** 10.3390/molecules28186648

**Published:** 2023-09-15

**Authors:** Krystian Klimczak, Monika Cioch-Skoneczny, Aleksandra Duda-Chodak

**Affiliations:** Department of Fermentation Technology and Microbiology, Faculty of Food Technology, University of Agriculture in Krakow, Al. Mickiewicza 21, 31-120 Kraków, Poland

**Keywords:** beer, dry-hopping, hop resins, essential oils, hop creep

## Abstract

Dry-hopping is the addition of hops to the wort on the cold side of the brewing process. Unlike standard hop additions, its main purpose is not to produce a characteristic bitterness but to extract as much of the hop essential oils as possible, which are largely lost in the standard hopping process. When dry-hopped, it is possible to obtain a beer with an aroma that is difficult to achieve when hops are used on the hot side of the brewing process. As a result, this process has become very popular in recent years, particularly in beers that belong to the ‘craft beer revolution’ trend. In addition, the usefulness of this process is increasing with the development of new hop varieties with unique aromas. This article presents the main components of hops, focusing on those extracted during the process. Changes in the composition of beer bittering compounds and essential oils resulting from this process are discussed. This paper presents the current state of the knowledge on the factors affecting the degree of extraction, such as hop dosage, the time, and temperature of the process. Issues such as process-related physicochemical changes, hop creep, low flavor stability, haze formation, and green flavor are also discussed.

## 1. Introduction

Brewers use a variety of methods to achieve a product with the desired sensory characteristics. One such method is dry-hopping. This is a process in which hops are added on the cold side of the brewing process. Typically, the hops are added after the primary fermentation, but this is not required. In some cases, the hops are added earlier in the process. This process makes it possible to produce beer with sensory characteristics that are difficult to obtain when hops are used at earlier stages of the wort production. When the hops are added during the ‘hot part of the production process’, their main role is to ensure an appropriate level of bitterness. However, the environmental conditions during the wort boiling process result in the loss of a substantial part of the hop essential oils. This occurs through phenomena such as evaporation or oxidation of essential oil components. Brewers attempt to counteract this loss by adding aroma hops in the final stages of boiling (typically within the last 15 min of the boiling), after the flameout, or during the whirlpool. The shorter boiling time allows more of the essential oils to be retained without adding excessive bitterness (due to the reduced time in which α-acid isomerization takes place). However, the highest degree of essential oil preservation is achieved when hops are added during the cold part of the beer production process (as the evaporation is greatly minimized). This procedure, combined with the use of novel or experimental hop varieties, makes it possible to produce a beer with unique aromas, such as fruity, floral, or resinous. However, it can also produce other changes in the sensory characteristics, such as modifications of the bitterness level or its quality. Thus, this seemingly simple process causes many changes in the physicochemical parameters and sensory characteristics of the beer. For a long time, the changes that occur during dry-hopping were not fully understood scientifically. In recent years, however, a number of articles have been published that allow for a more complete understanding of this issue.

Dry-hopping involves adding hops as pellets or dry hop cones. The process can be categorized into two types: static and dynamic dry-hopping. Static methods rely on a simple addition of hop cones/pellets into the fermenter. This approach is predominantly used in small breweries and home brewing. On the other hand, dynamic dry-hopping is becoming increasingly popular in many breweries. It involves multiple techniques (such as stirring beer or pumping it through a layer of hops) to suspend the hop particles in beer. It significantly shortens the process time and facilitates the transfer of hop compounds into a beer. The standard hop dosage used in this process for beers produced in the USA is in the range of 500–800 g/hL. However, there is considerable variability in dosage, depending on the intended sensory impact of the process. In some cases, extremely high doses of 2200 g/hL are used, which is often unnecessary and not economically justified. It should be noted that hops are the most expensive of the raw materials typically used in brewing. The previously mentioned range (500–800 g/hL) is the optimal dry-hopping dose, according to Lafontaine and Shellhammer [1]. Above the aforementioned range, the efficiency of dry-hopping noticeably decreases due to the saturation of the environment with hop oils. Extremely high hop doses do not seem to have an adequate impact on the sensory characteristics of the product, contrary to the cost of its production. Nevertheless, it is possible to find beers on the market in which very high hop doses are used [1,2,3].

As with many aspects of brewing, it is difficult to pinpoint when dry-hopping was invented and who was the first brewer to use this method. Most papers on the history of dry-hopping suggest that it was first used by British brewers in the 18th and 19th centuries. The most commonly cited hypothesis is that it was invented by George Hodgson of the Bow Brewery located in East London. Some sources credit him with developing the India Pale Ale style. According to this theory, Hodgson began exporting a variation of his Pale Ale (available to Londoners) to India in the 1790s. Barrels destined for the long ocean voyage were enriched with a significant addition of hop cones. This addition was intended to improve the sensory characteristics of the beverage as it endured the long voyage and to reduce spoilage. Interestingly, although this theory has been embedded in the culture, it probably has little to do with reality. The historical sources supporting this theory are rather circumstantial. Some sources suggest that dry-hopping may have been used in earlier years [4,5]. Nowadays, this procedure has gained a widespread appeal, especially in the beers that can be attributed to the craft beer revolution. It has become a standard method in the brewing industry, where it is mainly associated with top-fermented beers, but it has also been used in lager beers.

The aim of this review is to present the current state of knowledge regarding the changes in the physicochemical parameters and sensory attributes of beers resulting from dry-hopping. The article also attempts to describe the factors influencing the efficiency of the process, the stability of the characteristics imparted by this process, and the problems encountered when dry-hopping is used.

## 2. Chemical Composition of Hops

Today, hops are considered as an essential ingredient in beer. They give a beer its specific taste and aroma. From the brewer’s point of view, the most important components of hops are the hop resins, which give the beer its characteristic bitterness, and the essential oils, which are responsible for the aroma. Table 1 shows the groups of compounds found in hops and their typical contents. It should be noted that the composition of hops depends on many factors. The first of these is primarily the hop variety. More than 260 hop varieties are currently available, varying in their content of resins (especially alpha acids, a component of resins), essential oils, and polyphenols. But most importantly, the sensory qualities can be very different from one variety to another [6]. In addition, the content of hop compounds is influenced by factors such as growing conditions, processing, and storage. The proportion of individual compounds in the aforementioned groups also varies. The chemical composition of hop varieties determines the different perceptions of bitterness resulting from the use of distinct varieties. It should also be noted that, according to Hanke et al. [7], the content of α-acids and essential oil does not show a clear correlation with the actual content of flavor compounds. Lafontaine et al. [1] investigated the effect of the harvest date on the sensory characteristics and essential oil content of Cascade hops. In the authors’ study, hops harvested later in their technological maturity were more suitable for dry-hopping. These hops produced a beer with more citrus sensory characteristics. The authors attributed this observation to the higher content of volatile essential oil fractions, including volatile thiols (as the hops mature, the proportion of bound polyfunctional thiols decreases in favor of the free form fraction) [8,9,10,11].

**Table 1 molecules-28-06648-t001:** Typical chemical composition of freshly dried hop cones [8,12].

Compound Group	Typical Content (% (*w*/*w*))
Resins	15–30
Essential oils	0.5–3
Proteins	15
Monosaccharides	2
Polyphenols	4.3–14
Pectins	2
Amino acids	0.1
Waxes and steroids	trace-25
Ash	8
Moisture	10
Cellulose and others	40–43

### 2.1. Hop Resins

From a technological point of view, hop resins are the components responsible for the valued bittering properties of hops. According to Almaguer et al. [8], hop resins can be divided into soft resins (which include α- and β-acids); hard resins (α, β, δ, ε, and the uncharacterized hard resin fraction—about 2–3% of the dry weight of the hop cone); and the intermediate oxidation products of soft resins (humulinones and hulupones). Although hop resins are characterized by a large number of different groups of compounds, with the exception of α-acids, most of them have been relatively little studied. This may be due to the fact that α-acids, which are isomerized to forms of iso-α-acids during the boiling process, are the main factor responsible for bitterness. The effect of hard resins on beer bitterness is probably small, according to the current literature.

The varieties evaluated by Baker et al. [13] contained α-acids in the range of 1.3% to 12.6% of the dry hop cone weight, depending on the variety. Today, it is easy to find super-bittering hop varieties on the market that contain more than 15% (*w*/*w*) of these compounds. An example of such a variety is Columbus, which contains 14–18% (*w*/*w*) α-acids. The conditions during the dry-hopping process do not allow the isomerization process to take place, so they remain in a form that does not affect the taste of the beverage. Alpha acids are poorly soluble in water. However, according to Maye et al. [14], they can be used as foam-stabilizing agents. The often-quoted solubility limit for α-acids in finished beer is about 14 mg/L. According to Fritsch and Shellhammer [15], α-acids, even at a concentration of 28 mg/L (which, according to the authors, is close to the solubility limit of these compounds in lager beer), do not impart a noticeable bitterness to the beverage. According to Maye and Smith [16], the typical concentration of α-acids (determined using HPLC) in dry-hopped IPA is ~13 mg/L. When examining New England India Pale Ale (NEIPA)-style beers, the authors found concentrations as high as 72 mg/L. Other factors, discussed later, are likely responsible for such high concentrations, and most beers have relatively low α-acid levels [6,14,15,16].

The other group of compounds, β-acids, should also be mentioned. According to Baker et al. [13], these compounds make up 1.0–6.8% of the dry hop cone weight, depending on the variety. Other authors have presented even larger ranges. Due to the very low solubility of β-acids in water, it has long been believed that compounds from this group do not have a significant effect on beer flavor. About 85% of these compounds remain in the hops after boiling. However, β-acids can undergo transformations during boiling. The products of these reactions, characterized by a greater solubility in water, have some influence on the sensory characteristics of the beverage. Hulupones are an example of such products. It is currently believed that the unreacted forms of β-acids do not have a significant effect on the sensory characteristics of beer. To the best of the authors’ knowledge, this issue has not been thoroughly investigated. However, it is known that these compounds are transferred into beer during dry-hopping and can be detected in such beers. β-acids exhibit a strong antimicrobial activity, so their extraction may have a beneficial effect on the microbiological stability of beer [13,17,18,19].

Humulinones and hulupones are oxidation products of α- and β-acids, respectively (Figure 1) [20]. The available literature shows that humulinones are the main factors responsible for the increase in perceived bitterness in beers after the dry-hopping process. Their content in the dry hop cone is usually less than 0.5%, and they are not detectable in fresh hops. Their content increases during hop storage. Algazzali and Shellhammer [21] conducted a sensory evaluation with a panel of nine individuals, all of whom had received prior training in describing bitterness quality and conducting descriptive analyses. They also received additional training sessions. The panelists assessed samples of beers enriched with the studied hop acids on a 0–15 point scale. The humulinones and hulupones were reported to be 66% (±13%) and 84% (±10%), respectively, as bitter as iso-α-acids. The compounds from this group are commonly detected in dry-hopped beers, as shown in the study by Hahn et al. [22], where the average humulinone content measured for 121 commercially available beers was 17 mg/L, and a detectible level (≥1 mg/L) was found in 117 of the beers tested (determined by a HPLC analysis). The authors noted that hulupones were also detected in a small number of beers but below the limit of quantification (the authors, however, did not state the LQQ value) [8,13,15,21,22,23,24,25].

**Figure 1 molecules-28-06648-f001:**
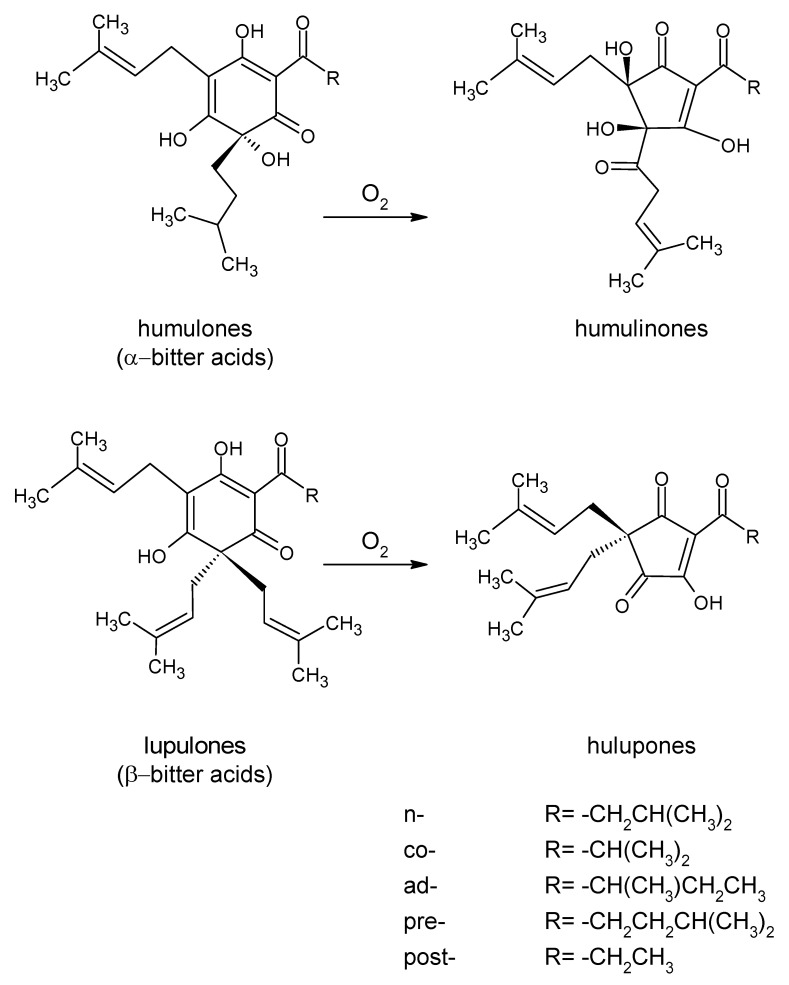
Schema of the oxidation of α-bitter acids and β-bitter acids to humulinones and hulupones, respectively [20,26]. Prefixes n-, co-, ad-, pre-, and post- are identical for various analogs of both humulones and lupulones and are determined by the structure of the acyl side chain (R), which is presented below the figure.

The currently available literature suggests that components other than α-acids, humulinones, hulupones, and polyphenols do not have a significant effect on the sensory characteristics of beer when dry-hopped. Alternatively, their effect has not been sufficiently studied, so they are not included in this article. Note, however, that if hops are used during the hot part of the production process, it is possible to obtain sensory-active compounds from other components. An example is the previously mentioned β-acid reactions occurring under wort boiling conditions [27].

### 2.2. Essential Oils

Essential oils are responsible for the characteristic aromas of the hop varieties, as well as the flavor they impart to the beer. The dry-hopping process allows their influence on the sensory qualities of the beverage to be maximized. The most commonly quoted content of essential oils in the dry hop cone is 0.5–3%; however, some sources state that their content can be as high as 4%. Interestingly, the division between aroma and bittering hops does not always correspond to the essential oil content. In a study by Aberl and Coelhan [28], bittering hops often had a higher essential oil content than aroma hops, as determined using Headspace Trap GC-MS and steam distillation based on the EBC and ASBC methodologies.

The essential oil content cannot be used to differentiate between most cultivars [28,29,30]. Studies suggest that the number of individual compounds in this fraction may be close to 1000 [29,31]. Sharpe and Laws [32] divided the compounds found in the essential oil fraction into three groups: hydrocarbons, oxygen-containing compounds, and sulfur compounds. The structures of the most important compounds present in hops are shown in Figure 2, Figure 3 and Figure 4.

From a chemical point of view, hydrocarbons constitute the majority of hop essential oils (50 to 80% of the fraction). Due to the nonpolar nature of the compounds in this group, they are found in significant amounts primarily in dry-hopped beers. The content of individual hydrocarbons in particular hop varieties is characterized by high variability and may therefore be a distinguishing characteristic of specific hop varieties. Nance and Setzer [33] studied the chemical composition of seven commonly used hop varieties using GC-MS. The evaluated hops were hydrodistilled for 4 h with continuous extraction with dichloromethane. The percentages of the components were calculated based on the total ion current without standardization. In their study, the content of monoterpene hydrocarbons in essential oils ranged from 9.4 to 52%, while sesquiterpene hydrocarbons ranged from 29.1 to 70%. Although it is often stated in the literature that the main hydrocarbon found in hops is β-myrcene (**1**) (Figure 2), this is not always true. This compound was predominant in Cascade (48.9% of the hop essential oil), Northern Brewer (52.4%), Saaz (25.3–25.7%), and Wilamette (40.9%), while α-humulene (**29**) dominated in Hallertauer (22.7–28.0%), Vanguard (51.2%), and Sterling (41.6%) hops [28,32,33,34,35].

The second group, accounting for about 30% of hop essential oils, are oxygen-containing compounds, which include a variety of compounds belonging to the alcohols, esters, acids, ketones, aldehydes, lactones, and epoxides (Figure 2, Figure 3 and Figure 4). They impart mostly floral and fruity aromas. Among the alcohols, the terpene alcohol linalool (**3**) is the most abundant in most varieties. Linalool (**3**) is a product of β-myrcene (**1**) hydration. Other terpene alcohols found in hops are geraniol (**4**), nerol (**5**), terpineol (**7**) (Figure 2), and nerolidol (**31**) (Figure 4). They are present in smaller amounts than linalool (**3**); however, some varieties may be rich in geraniol (**4**). Specific terpene alcohols can be present both as free molecules and bound to carbohydrates. Some of these compounds are believed to have a significant influence on the aroma of dry-hopped beers. They are thought to be at least partially responsible for the characteristic citrus/floral aroma of such beers. Some sources suggest that synergistic effects between compounds may be an important factor in beverage aroma perception. For example, according to Takoi et al. [36], the perceptibility of geraniol (**4**) may be enhanced by the presence of linalool (**3**). Similar to hydrocarbons, there is considerable variation in the content of these compounds between varieties. The research suggests that the geraniol (**4**) content, in particular, varies considerably between varieties. The average content of terpene alcohols in hops is in the range of 4000–8500 mg/kg. The average content of aliphatic alcohols in hop varieties studied by Nance and Setzer [33] was in the range of 0.0–1.1% of essential oil. 

**Figure 2 molecules-28-06648-f002:**
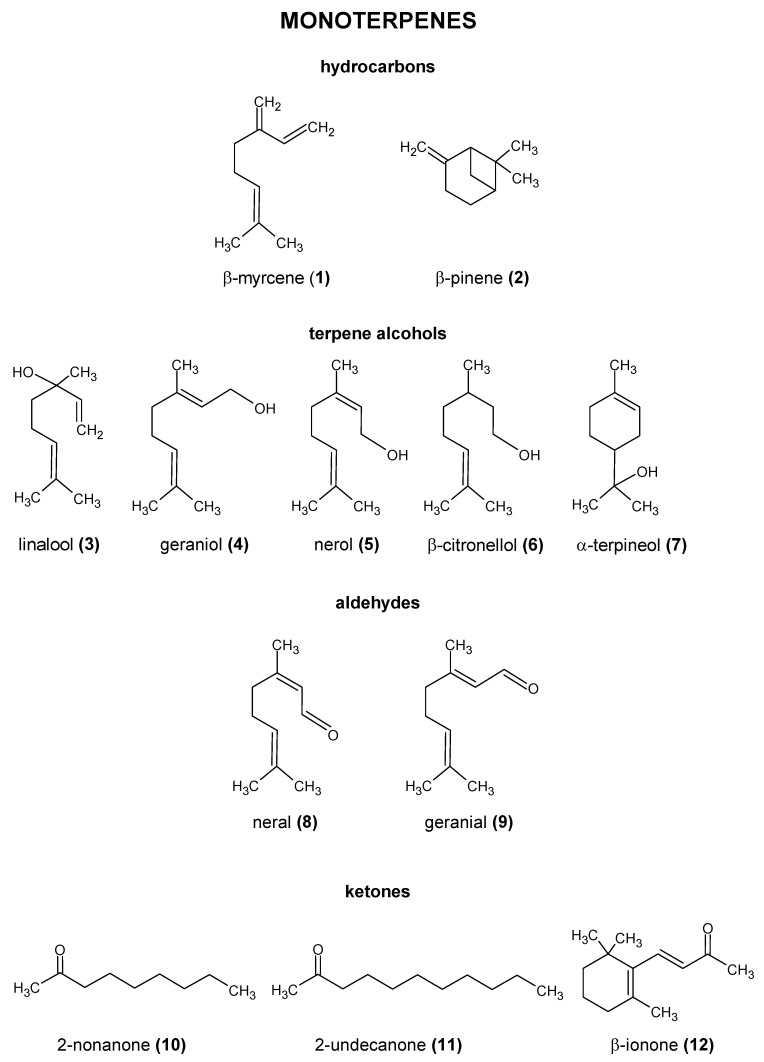
Chemical structure of the most important compounds present in hops [26,37,38,39,40].

**Figure 3 molecules-28-06648-f003:**
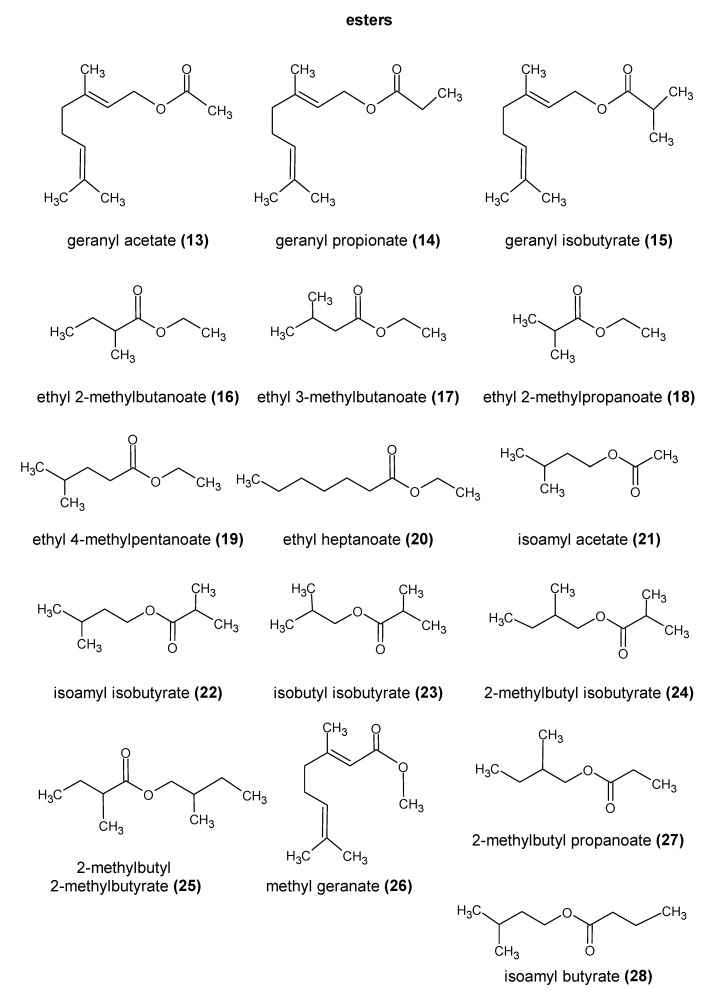
Chemical structure of the most important esters present in hops [40] PubChem database.

In recent years, the bound forms of terpene alcohols have attracted considerable interest. The release of terpene alcohols from their glycosides could intensify the aromas imparted by these compounds (generally referred to as citrusy). Lafontaine et al. [41] found that the content of glycosidic compounds is mainly influenced by the hop variety and secondarily by the degree of plant maturity. Cibaka et al. [42] investigated the potential of glycosidically bound fractions of terpene alcohols to influence beer aroma, and five distinct hop varieties were assessed. The compounds studied were extracted from dry-hopped beer and treated with a commercially available β-glucosidase. The use of β-glucosidase released linalool (**3**), α-terpineol (**7**), β-citronellol (**6**), and geraniol (**4**) in extracts from all the evaluated beers (hopped with five hop varieties). However, the amounts obtained were much lower than the compounds already present in the beer (0.6 –28.6 mg/kg of aglycons compared to 7.8–109.2 mg/kg of free forms). In hops studied by Lafontaine et al. [41], linalool (**3**) and α-terpineol (**7**) were mainly present in bound form. The estimated content of these compounds in the hop varieties studied was in the range of 0.46–1.81 µg/g of 100% dry hops (calculated in relation to the internal standard used—decyl-β-D-glucopyranoside). The authors also noted that these glycosides are mainly found in the form of pentose-hexose-monoterpenols. Previously, it was believed that they occur in the form of hexose-monoterpenols. It is commonly mentioned in the literature that glycosides of monoterpene alcohols can be released through the enzymatic activity of yeast β-glucosidase and β-glucanase. However, Lafontaine et al. [41] suggested that, due to the different form of the glycosides (hexose-pentose-monoterpenols instead of hexose-monoterpenols), other activities, such as α-L-arabinofuranosidase and/or α-L-rhamnosidase, may be involved in the release of these compounds. These results may indicate that the aromatic potential of hop glycosides is not as high as previously thought [36,41,42,43,44,45,46].

An important component of plant essential oils, including hops, are compounds that can be classified as esters. According to the literature, hop essential oil can contain up to 15% of these compounds. In a study by Nance and Setzer [33], the essential oil of the hop varieties evaluated contained 1.2–9.1% of the carboxylic esters. They impart floral, fruity, and, sometimes, solvent aromas. Among the esters found in hop oil, a homologous series of methyl esters from hexanoate to dodecanoate are the most prominent. Branched-chain and unsaturated methyl esters are also quantitatively important groups. It is likely that methyl esters derived from hops undergo transesterification reactions to form corresponding ethyl esters, releasing methanol in the process. Another important group of compounds are the geranyl esters (e.g., geranyl acetate (**13**)**,** geranyl propionate (**14**), and geranyl isobutyrate (**15**)). These are likely to be hydrolyzed into geraniol (**4**). Thus, the transfer rates of geraniol may be greater than 100% of its free form content in dry hop matter. Esters derived from hops, similar to hydrocarbons, are mostly found in dry-hopped beers (Table 2). This is mainly due to the high volatility and reactivity of this group of compounds [30,34,35].

Compounds such as aldehydes, ketones, epoxides, and lactones, although present in hops and often characterized by a low sensory threshold, have been relatively little studied. According to the older literature, epoxides and diepoxides derived from hops undergo hydrolysis and various rearrangements during the beer production process. As a result, they form the corresponding ketones and alcohols. The aldehydes found in hops, such as neral (**8**) and geranial (**9**), undergo similar transformations, where they are reduced to their corresponding alcohols. However, more recent studies on epoxides indicate that they can also have a significant impact on beer aroma. Praet et al. [2] reported that compounds such as humulene epoxide I (**32**) and II (**33**) (Figure 4) and α-humulene-derived epoxides may be important determinants of kettle-hopped beer aroma. Humulene epoxide I is relatively resistant to hydrolysis and is found at concentrations well above its sensory threshold in kettle-hopped beers. These compounds are also found in dry-hopped beers. The fatty acids found in hops are usually associated with the degradation products of α- and β-acids and are found in higher amounts in older hops. However, according to Olšovská et al. [47], malt is a much more important source of these compounds than hops. At the same time, they may be precursors of other volatile compounds [2,29,47].

**Figure 4 molecules-28-06648-f004:**
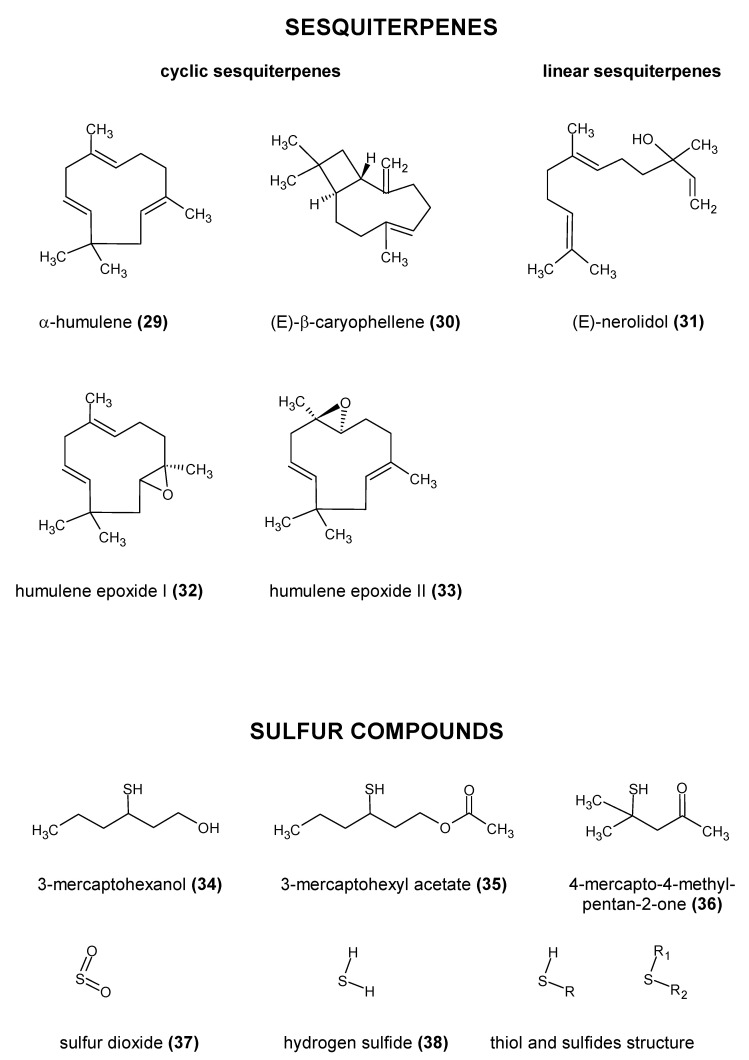
Chemical structure of the most important esters present in hops [37,38,39,40].

It should be highlighted that the levels of detected compounds in dry-hopped beer depend on the analytical method used. Trap GC-MS typically gives higher values than HS-GC-MS, as seen with α-humulene (**29**), β-myrcene (**1**), linalool (**3**), and geraniol (**4**). However, when SPME extraction is used in conjunction with HS-GC-MS, the concentration of the compound of interest may be even higher. Of course, utilizing the same method does not necessarily yield identical outcomes, since the hop variety also has an influence.

**Table 2 molecules-28-06648-t002:** Selected compounds detected in dry-hopped beers [48,49,50].

**Compound**	**Aroma**	**Aroma** **Threshold**	**Example of Concentrations** **(±SD *) in Dry-Hopped Beer (the Methods Used for Analysis)**	**Reference**
**Hydrocarbons**				
α-humulene (**29**)	spicy and woody	450 µg/L, 120 µg/L	5.2 (±0.5) µg/L (Trap GC-MS) 0.4 (±0.04)–1.2 (±0.41) μg/L (HS-GC-MS)	[51,52,53,54]
β-myrcene (**1**)	herbaceous, resinous, green, balsamic, fresh hops	350 µg/L, 30–200 µg/L, 9.5 µg/L	79.7 (±2.8) μg/L (Trap GC-MS) 0.3 (±0.12)–15.9 (±2.58) μg/L (HS-GC-MS) 117.4–863.6 μg/L (HS-SPME-GC-MS)	[51,52,53,54,55,56]
β-pinene (**2**)	turpentine odor with a dry, woody, or resinous aroma	140 µg/L	15.4 (±3.3)–89.4 (±19.7) μg/L (HS-GC-SPME)	[50,57,58]
(E)-caryophyllene (**30**)	spicy and woody	230 µg/L	2.3 (±0.2) μg/L (Trap GC-MS) 0.2 (±0.06)–0.3 (±0.12) μg/L (SPE-GC-MS)	[51,52,53,54]
**Terpene Alcohols**				
α-terpineol (**7**)	lilac odor with a sweet taste reminiscent of peach on dilution	330 μg/L, 450 μg/L	25.4 (±6.4) μg/L (Trap GC-MS) 0.2 (±0.07)–2.8 (±0.15) μg/L (HS-GC-MS)	[36,54,55,57,59]
β-citronellol (**6**)	lemon/lime-like	8 μg/L, 40 μg/L	26.3 (±3.5) μg/L (Trap GC-MS) 1.2 (±0.05)–1.8 (±0.29) μg/L	[36,42,54,55]
geraniol (**4**)	floral/rose-like	4–5 μg/L, 40 μg/L	265 (±45.8) μg/L (Trap GC-MS) 30.9 (±0.92)–72.8 (±0.17) μg/L (HS-GC-MS)	[36,41,54,55]
linalool (**3**)	floral/lavender/coriander/citrus-like flavor	8 μg/L	155 (±8.0) μg/L (Trap GC-MS) 33.4 (±0.80)–36.4 (±0.87) μg/L (HS-GC-MS)	[42,54,55]
nerol (**5**)	fresh, sweet, rose-like	300 μg/L, 80 μg/L	5.2 (±0.10)–7.7 (±0.25) μg/L (SPME-GC-MS)	[36,57,59]
**Sulfur Compounds**				
3-mercaptohexanol (3MH) (**34**)	rhubarb and grapefruit	55 ng/L	29–475 ng/L (LC-MS/MS)	[55,60]
3-mercaptohexyl acetate (3MHA) (**35**)	passion fruit and grapefruit	4 ng/L	4–10 ng/L (LC-MS/MS)	[55,60]
4-mercapto-4-methylpentan-2-one (4MMP) (**36**)	blackcurrant and passion-fruit-like	1.5 ng/L	31–40 ng/L (LC-MS/MS)	[55,60]
**Esters**				
ethyl 2-methyl- butanoate (**16**)	green-fruity, apple-like	1.1 μg/L	1.1–1.9 μg/L (SPME-GC-MS)	[56,57]
ethyl 3-methyl- butanoate (**17**)	fruity, vinous, apple-like	2 μg/L	1.5–1.8 μg/L (SPME-GC-MS)	[56,57]
ethyl 2-methyl- propanoate (**18**)	fruity	6.3 μg/L	11.8–18.7 μg/L (SPME-GC-MS)	[56,57]
ethyl 4-methyl- pentanoate (**19**)	fruity	1.0 μg/L	0.6–0.9 μg/L (SPME-GC-MS)	[56,57]
ethyl heptanoate (**20**)	fruity, reminiscent of cognac, wine/brandy	400 μg/L	1.6–8.8 μg/L (SPME-GC-MS)	[57,61,62]
isoamyl acetate (**21**)	fruity, banana, sweet, fragrant	1200 μg/L	275 (±54.6)–(330 (±11.6) μg/L (Trap GC-MS)	[54,55,57,63]
isoamyl isobutyrate (**22**)	fruity, apricot, pineapple	N/A **	0.6–16.5 μg/L (SPME-GC-MS)	[57,62]
isobutyl isobutyrate (**23**)	pineapple	N/A	0.4–54 μg/L (SPME-GC-MS)	[57,61]
2-methylbutyl isobutyrate (2-methylbutyl 2-methylpropanoate) (**24**)	fruity	78 μg/L	1.6–103.6 μg/L (SPME-GC-MS) 41–198 μg/L (GC-FID) 24–87 µg/L (HS-SPME-GC-MS)	[57,61,63,64,65]
2-methylbutyl 2-methylbutyrate (**25**)	fruity, floral, banana and pineapple	N/A	0.13–0.3 μg/L (SPME-GC-MS)	[66]
methyl geranate (**26**)	fruity, floral, waxy, herbal, citrus/sweet, candy	N/A	1.3–7.6 µg/L (SPME-GC-MS) 8–192 µg/L (HS-SPME-GC-MS)	[1,65,66]
2-methylbutylpropanoate (**27**)	sweet, fruity, rum-like	N/A	41–198 µg/L (SPME-GC-MS)	[66]
isoamyl butyrate (**28**)	fruity, green apple-like, and apricot	N/A	N/A	[66]
**Ketones**				
2-nonanone (**10**)	rue, rose and tea-like	5–200 μg/L	6.9 (±0.30) μg/L (Trap GC-MS) 0.1–2.7 µg/L (SPME-GC-MS)	[54,56,57]
2-undecanone (**11**)	rue odor with a sweet flavor reminiscent of peach	7 μg/L, 400 μg/L	4.4 (±0.30) μg/L (Trap GC-MS) 0.1–6.0 µg/L (SPME-GC-MS)	[54,56,57]
β-ionone (**12**)	violet-like, fruity, woody	N/A	≤0.1 μg/L (SPME-GC-MS)	[57,61]

* Standard deviations presented in the parentheses where stated by the authors. ** N/A—no available data and n.d.—not detected.

The last important group of compounds are the sulfur compounds; among which, the polyfunctional thiols have attracted particular attention in recent years. Of the volatile sulfur compounds, sulfur dioxide (**37**) (SO_2_), hydrogen sulfide (**38**) (H_2_S), thiols (R–SH), and sulfides (R–S–R) are the most abundant (Figure 4). In some cases, hops may contain sulfur compounds derived from plant protection products. The content of polyfunctional thiols in hops, as determined by Cibaka et al. [44] using GC-PFPD, was in the range of 0.015–1.296 mg/kg of hops. On the basis of these results, the content of these compounds in hops can be described as insignificant. However, the human olfactory system is highly sensitive to sulfur compounds. They can significantly affect the aroma and the sensory characteristics of beer, even at very low concentrations. For some of these compounds, concentrations exceeding 1.5 ng/L (Table 2) are already detectable in the final product. The occurrence of these compounds in hops has attracted widespread attention in recent years. It has been suggested that they may be responsible for the tropical flavors obtained with certain hop varieties, where they are known to impart aromas such as guava, passion fruit, and mango. It is now suspected that, for some hop varieties, they may be, together with monoterpene alcohols, important determinants of the beer aroma obtained when dry-hopping is used. Polyfunctional thiols can occur both in free form and as glycine or cysteine conjugates. Glutathionylated precursors were the major form of polyfunctional thiols in the hop varieties studied by Cibaka et al. [67] (Amarillo, Hellertau Blanc, and Mosaic). In the hop varieties tested, the S-cysteine-bound fraction accounted for 23 to 126 times more of these compounds than the free form fraction. The authors also found that the contents of these compounds varied significantly between hop varieties, which may be responsible for the differences in their aroma [67]. The individual varieties were also characterized by different ratios of the individual bound forms to the volatile forms. Yeast β-lyases can release cysteine-bound polyfunctional thiols during fermentation. In the case of gluthationated precursors, progress in understanding the mechanism of their release has been made by Cordente et al. [68]. It should be noted that malt is also likely to be an important source of these precursors. This was confirmed by the results of Chenot et al. [69]. Currently, most studies focus on 3-mercaptohexanol (**34**) (3MH), 4-mercapto-4-methylpentan-2-one (**36**) (4MMP), and 3-mercaptohexyl acetate (**35**) (3MHA) [29,30,44,67,68,69,70,71,72].

### 2.3. Polyphenols

It is also important to discuss the polyphenol content in the dry hop cone. These compounds are commonly attributed with a role in the antioxidant activity of beer or turbidity formation. Zhao et al. [73] studied 40 lager beers and observed a correlation, where the antioxidant activity of the beer increasing as the total polyphenol content (determined according to the Folin–Ciocalteu spectrophotometric method) increased. The literature on dry-hopping may suggest that this process can also significantly alter the taste of beer, imparting astringency and bitterness. Polyphenols extracted during dry-hopping may be responsible for some of these sensations. McLaughlin et al. [74] conducted a study in which a certain amount of polyphenols (extracted from spent hop powder after CO_2_ extraction) and/or iso-α-acids were added to lager beer. Beers to which polyphenols had been added were characterized by higher levels of bitterness, which was described as lingering. In addition, they received higher scores in the categories of ‘metallicity’, ‘sharpness’, and ‘medicinal’ flavor. According to Goiris et al. [75], various fractions of polyphenols are characterized by different effects on taste, ranging from imparting fullness of flavor to astringency. This issue was further explored by Gribkova et al. [76]. The authors suggested that a combination of bitter resins, prenylflavonoids, and catechins is probably responsible for sharp astringency. Catechin, quercetin, and rutin, associated with soluble nitrogen and β-glucan dextrins, are at least partially responsible for residual bitterness [73,74,76,77].

## 3. Extraction of Compounds during the Process

From the moment the hops are added to the wort/beer during fermentation, the extraction process begins. It is influenced by a whole range of factors that will be discussed later in this article. It should be noted that different groups of compounds are extracted in different proportions. In a study by Hauser et al. [17], hops used for 24-h dry-hopping contained 33–51% of their initial essential oil content. The value was determined according to hydrodistillation ABSC method hops 13, with a subsequent GC-MS analysis. Much of this loss was due to the extraction of oxygenated compounds (such as terpene alcohols). Their levels dropped to 10% of their initial content. Hydrocarbons were largely retained in the spent hops. In the case of α-acids, their content decreased to 52–77% of the original amount, as determined using HPLC. However, in the study of Gasiński et al. [78], the bittering potential of hops used in dry-hopping, if used in another brew on the hot side of the process, decreased only by 15–30%, as assessed by a spectrophotometric IBU analysis at λ = 275 nm. Overall, polar compounds (such as linalool (**3**)) are transferred in large amounts, while nonpolar compounds (such as β-myrcene (**1**)) are found in much smaller amounts due to their low solubility, adsorption on the yeast cell surface, and volatilization with carbon dioxide [17,78].

### 3.1. Volatile Compounds

In general, the degree of extraction of aroma compounds during the dry-hopping process depends mainly on the nature of the compound, i.e., its polarity. Hydrocarbons, although representing a significant portion of hop essential oil, are extracted in relatively small quantities due to their high hydrophobicity. In contrast, the extraction rate of polar compounds is at least high, according to all the sources collected. Haslbeck et al. [79] estimated the extraction rate of C11 esters, monosesquiterpenes, and sesquiterpenes at 0.1–13%, with alcohols and C8 esters at more than 23%. The authors used GC-FID to analyze the compositions of the essential oils of hops used and HS-GC-MS to analyze the beer volatiles. The very low extraction rate of hydrocarbons such as β-myrcene (**1**), β-caryophyllene (**30**), α-humulene (**29**), and β-farnesene was also confirmed by Lafontaine and Shellhammer [1] (using static dry-hopping, the GC-MS analysis for hop oils, and SPME-GC-MS for beer volatiles) and Forster and Gahr [49] (distillation and GC-FID and SPME-GC-MS, respectively).

In the case of the representatives of terpene alcohols (linalool (**3**) and geraniol (**4**)), Haslbeck et al. [79] obtained transfer rates exceeding 100% of their original hop content. Similarly high extraction rates at hop doses of 40–103 g/hL were reported by Forster and Gahr [49], whereby 100% of linalool was extracted, while, in the case of geraniol, the transfer rates ranged from 50 to more than 100%. The authors attributed this fact to the varietal differences in the transfer rate of geraniol. Much lower extraction efficiencies of linalool and geraniol were obtained by Lafontaine and Shellhammer [1]: ~23% and ~13%, respectively. Extraction rates exceeding 100% can be explained by several phenomena. Such phenomena could be the biosynthesis of these compounds by yeasts or their formation during broadly defined biotransformation reactions. In a study by Takoi et al. [61], the concentrations of the terpene alcohols β-ionone (**12**) and nerol (**5**) were below the sensory threshold in all the beer variants hopped with 19 different hop varieties, using SPME-GC-MS to quantify them. These results indicate that these compounds are unlikely to be important components of the dry-hopped beer bouquet. The high extraction rate of the esters evaluated was reported by Forster and Gahr [49], where the esters (isobutyl isobutyrate (**23**) and isoamyl propionate) had a transfer rate of 53–83%. The high volatility and instability of these compounds mean that they are often found at higher concentrations in dry-hopped beers.

This hypothesis was corroborated by the results of Takoi et al. [80] (SPME-GC-MS), where isobutyric esters were mainly found in such beers. Regarding other groups of compounds, including polyfunctional thiols, there is little information on their extraction rates so far. Wéber [81] determined the extraction rates of 3S4MPol (3-sulfanyl-4-methylpentan-1-ol) and 4S4MPone (4-sulfanyl-4-methylpentan-2-one) to be 40% and 65%, respectively. Kohles et al. [82], using GC-MS, found the transfer rate of 4-mercapto-4-methylpentan-2-one (**36**) (4MMP) to be 46–79%. Thus, these compounds seem to be extracted from hops to a high degree. In the case of other compounds, such as humulene epoxide I (**32**), 2-nonanone (**10**), and 2-undecanone (**11**), their extraction rates assessed by SPE-GC-MS were reported to be 30.2–59.7, 0–27.4, and 11.1–41.3%, respectively, and they depended on the hop variety [1,50,61,81,82,83].

### 3.2. Bitter Compounds

Dry-hopping can significantly affect the content of bittering agents in the beer. As the levels of these compounds change, the perceived bitterness, as well as the IBU (international bitterness units) levels can also change. It is important to note that bitterness is not a direct equivalent to IBU. In some cases, the value of IBU can increase disproportionately to the perceived bitterness level. This occurs as a result of the extraction of substances (from hops) that absorb similar wavelengths of light as iso-α-acids but do not have their bitterness. Hahn et al. [22] proposed an alternative method for evaluating the perceived bitterness. The aforementioned example method determines perceived bitterness by measuring the concentration of bitter components (e.g., humulinones and isohumulones) and other beer constituents (e.g., ethanol). However, other beer components such as polyphenols [84] or volatile compounds [85] are known to alter perceived bitterness. Hence, developing an analytical method that fully captures the perceived bitterness is a complex issue. A decrease in iso-α-acids, and a concomitant increase in α-acids, humulinones, and polyphenols, which are extracted from hops during the process, are well documented in the literature. Iso-α-acids from beer are adsorbed onto the hop mass, as was demonstrated by Oladokun et al. [11]. The question of whether other compounds found in hops, such as the ε-fraction found in hard resins (characterized by the solubility in aqueous ethanol solutions), are transferred into beer has not yet been investigated. If such extraction occurs, they could increase the level of perceived bitterness. To date, relatively little information has been published on the content of hulupones in beers that have been dry-hopped.

A significant change in the composition of the bitter compounds as a result of the dry-hopping process was shown by Maye and Smith [86] using HPLC analysis. The authors added Cascade hops at 1 lb./bbl. (1 pound per beer barrel is the equivalent of 386.54 g/hL) for 3 days at 16 °C. The initial content of 51 mg/L iso-α-acids decreased to 32 mg/L. The content of α-acids and humulinones increased from undetectable to 13 mg/L and 13 mg/L, respectively. Similar results were obtained by Forster and Gahr [49] (by HPLC). In their study, only 4–5% of the α-acids present in the hops were dissolved in the beer. At the same time, the decreases in α-acids in hops were much smaller than those reported by Hauser et al. [17]. This may be related to the solubility limit of α-acids in beer and the dose of hops used. Cocuzza et al. [24] reported the extraction of humulinones in the range of 41.4–69.2% of their content in the hops (using the EBC 9.50 HPLC method). Much higher extraction rates of humulinones (113% and 98%), using similar hop doses, were reported by Lafontaine and Shellhammer [1] and Smith et al. [87] (both using HPLC analysis), respectively. The significant impact of these compounds on the beer bitterness level was also emphasized by Ferreira et al. [88], who suggested that humulinones may account for up to 28% of the bitterness in dry-hopped beers. Oladokun et al. [11] and Maye and Smith [16] suggested even higher values, which depend on the beer style. Humulinones become a relevant determinant of the bitterness in dry-hopped beers mainly due to the significant doses of hops used in the process. In addition, humulinones are known to be lost at earlier stages of production. Their content is significantly reduced during boiling (22%) and fermentation (14%). According to Parkin and Shellhammer [89], a 7 mg/L increase in the content of these compounds increases the perceived bitterness of beer by 2.2 units on the 10-degree bitterness scale adopted by the authors. Oladokun et al. [11] indicated that humulinones may also be formed in beer during the dry-hopping process. This would explain the 113% humulinone extraction rate obtained by Lafontaine and Shellhammer [1]. During the dry-hopping process, β-acids are also extracted into the beer environment. Both α- and β-acids are extracted during the process and might be oxidized to some extent into humulinones and hulupones, respectively (Figure 1). However, this issue has not been thoroughly investigated. As mentioned previously, Hahn et al. [22] did not find significant amounts of hulupones in the beers they analyzed. This suggests that, if such reactions occur, their extent is probably minor [1,17,19,21,22,24,78,88,89,90].

### 3.3. Other Compounds

Hops contain a relatively significant proportion of polyphenolic compounds in their dry mass (Table 1). Polyphenols are largely extracted due to their hydrophilic nature. This was corroborated by the study of Forster and Gahr [49], where 50–60% of the total polyphenolic content of the hops used for dry-hopping were dissolved in a beer. It must be noted that the authors used a non-specific AHA method (AHA = Arbeitsgruppe Hopfenanalyse = Hop Analysis Working Group), coupled with spectrophotometric determination of the polyphenols according to EBC 9.11. This is significant, because, according to Parkin and Shellhammer [89], a 100 mg/L increase in beer’s polyphenol content is comparable to a 0.2 unit increase in perceived bitterness (on the 10-degree scale used in the authors’ study). The authors dry-hopped a beer with a baseline total polyphenol content of 111 (±5.1) mg/L using hop doses of 400 and 1600 g/hL for 72 h. The contents of these compounds increased to 185 (±1.6) mg/L and 211 (±2.8) mg/L, respectively. In the case of monoterpene alcohol glycosides, the hopping method does not appear to have a significant effect on the concentration of these compounds in the wort. A study by Sharp et al. [45] found no significant difference in the concentrations of these compounds between kettle, whirlpool, and dry-hopped beers. The authors used the hydrolysis of glycosides using a commercially available β-glucosidase and analyzed beer volatiles using HS-SPME GC–MS. This is the only study to date on this issue. However, as these compounds are composed of aglycones and sugar moieties, they should be characterized by a high degree of solubility. According to the current knowledge, glycosides are cleaved by the action of enzymes synthesized by yeast. Earlier hopping may likely allow a higher degree of glycoside cleavage. This is related to the relatively low activity of the enzymes responsible for these processes. Increasing the reaction time may allow higher amounts of volatile forms to be obtained. For the other compounds, no studies are available [45,89].

## 4. Parameters Affecting the Extraction Process

### 4.1. Hop Dose and Alpha Content

Lafontaine and Shellhammer [1] studied the effect of hop dosage on the extraction rate of hop-derived compounds and the resulting sensory qualities of the beer. The authors performed static dry-hopping for 24 h at 13.3 –15 °C using doses ranging from 0 to 1600 g hops/hL. Up to a dose of 800 g/hL, the two aroma descriptors studied by sensory analysis, citrus and herbal/tea, increased at similar, nonlinear rates. However, above this dosage, the process mainly imparted herbal/tea attributes. The content of compounds such as β-caryophyllene (**30**), α-humulene (**29**), terpinen-4-ol, α-terpineol (**7**), and geranial (**9**) increased nearly linearly with the increasing hop dose, while the extraction rate of terpene alcohols, i.e., linalool (**3**), geraniol (**4**), and nerol (**5**), decreased with the increasing hop dose. The extraction rates of the three mentioned terpene alcohols at a dose of 200 g/hL were ~23, ~13, and ~6%, respectively, while, at a dose of 1600 g/hL, they were only ~7%, ~3%, and ~1%, respectively. On the other hand, von Terzi [91] observed that the extraction rates of the polar compounds (esters and terpene alcohols) decreased more slowly with the increasing hop dose than that of the hydrocarbons. Overall, it seems that, for most of the hop-derived volatiles, higher hop doses resulted in lower extraction rates. The exception seems to be polyfunctional thiols, as Kohles et al. [82] did not observe a decrease in the extraction rates of these compounds with the increasing hop dosage. However, the authors used a dynamic dry-hopping method.

Regarding the bittering compounds, Lafontaine and Shellhammer [1] reported a similar decrease in the extraction rates depending on the hop doses used for the humulinones: 200 g/hL (113%), 386 g/hL (76%), 800 g/hL (74%), and 1600 g/hL (47%). These results were confirmed by the study of Maye et al. [92], although they obtained higher extraction rates at higher hop doses (88% humulinones dissolved at a dose of 773.1 g/hL). In the studies of Lafontaine and Shellhammer [1], the extraction rate of α-acid humulone was low and did not vary significantly with the dose used (386 g/hL (2%), 800 g/hL (1%), and 1600 g/hL (1%)). No significant changes were observed in the concentration of iso-α-acids in the beer as a function of the amount of hops used. Different results were obtained by Maye et al. [92], where the hop dosage used had a significant effect on both the increase in the α-acid content (in the hop range of 0 to 2319 g/hL) and the decrease in the iso-α-acid content (the largest changes occurred up to a dosage of 1160 g/hL). Maye et al. [92] used a similar process temperature (16 °C) but longer times (3 to 5 days), which might have had some impact on the obtained results. The results obtained by von Terzi [91] showed that bittering hops were characterized by a lower degree of essential oil extraction than varieties with a lower α-acid content (when the hop dose was based on the amount of essential oils). Other studies have reported similar results [1,50,91,92].

### 4.2. Time and Temperature

In the study by Oladokun et al. [11] (RP-HPLC), the greatest decrease in iso-α-acids content occurred during the first 24 h of the process, regardless of temperature (4 °C and 19 °C). It should be noted that the authors used dynamic dry-hopping, which significantly accelerates the extraction processes. At 4 °C, the loss of iso-α-acids continued at a much slower rate on subsequent days. The authors observed the greatest increase in the humulinone content within the first 24 h of the process. When highly bittering hops (Zeus) were used, temperature had no significant effect on the extraction rate of humulinones, and there was no significant change in their content after 3 days of the process. Different results were obtained for low-bittering hops (Hersbrucker), where higher humulinone contents were observed at the lower temperatures of the process. A significant increase in the humulinone concentration was also observed between days 10 and 14 for both temperatures tested. Overall, the final beers were characterized by a higher content of humulinones when hops with a higher α-acid content were used, and dry-hopping was conducted at a higher temperature. A similar relationship was found for α-acids, i.e., higher temperatures, as well as a higher content of these compounds in the hops, increased their content in the finished beers. Mitter and Cocuzza [93] confirmed (using HPLC) the effect of higher temperatures on higher extraction levels of α-acids. In addition, the authors pointed out that most of α-acids were extracted relatively quickly, i.e., α-acids concentrations of 2 mg/L were already obtained after the first day of the process. By the 18th day of hopping, their concentration had increased by only another 1 to 2 mg/L. Oladokun et al. [11] reported that higher process temperatures increased the extraction rate of hop polyphenols. The polyphenols were evaluated as the total polyphenols using the Folin–Ciocalteu method. This effect was particularly evident for the low-bittering hops. The authors explained this fact by the higher polyphenol content of these hops. Titus et al. [94] observed that the highest levels of the total polyphenol content among all the dry-hopped beer samples were found 3 h after the addition of the hops; after which, their content decreased. The content of these compounds did not return to this level even after 36 h of the process. These results were similar to those obtained by Oladokun et al. [11]. However, it should be noted that the analyses of Titus et al. [94] were characterized by much more frequent sampling during the initial hopping period than those of Oladokun et al. [11]. Also, Titus et al. [94] evaluated selected polyphenols (such as gallic acid, protocatechuic acid, p-hydroxybenzoic acid, vanillic acid, chlorogenic acid, sinapic acid, caffeic acid, p-coumaric acid, ferulic acid, (+)-catechin, (−)-epicatechin, (+)-catechin gallate, (−)-epicatechin gallate, quercetin-glucoside, and quercetin) by RP-HPLC-MS/MS and calculated the total polyphenol content by summing the assessed polyphenols. Thus, these results are not directly comparable. According to Lafontaine and Shellhammer [1], when static dry-hopping is used, 24 h is sufficient to extract an average of 75% of the humulinones at a process temperature of 13.3 to 15.0 °C.

When dry-hopping is used, brewers are often more interested in the aroma that can be achieved with this method. As previously mentioned, although β-myrcene (**1**) is one of the most abundant compounds in hops, its concentration in beers is relatively low. It has been suspected that it is due to its insolubility, evaporation, or adsorption (for example, on the foam). However, this problem was explained by Haslbeck et al. [55]. The authors evaluated the extraction rates of hop aroma compounds, along with the addition of yeast cells. In this study, the greatest losses of β-myrcene (**1**) occurred through evaporation of this compound with volatilized CO_2_, as well as absorption on the yeast cell surface. Higher temperatures (22 °C vs. 8 °C) increased the evaporation of this compound. The authors reported that more polar compounds such as linalool did not adsorb significantly onto the yeast cell surface. Interestingly, when the authors used a modification to prevent evaporation of this compound, 98–99% of β-myrcene (**1**) was adsorbed onto yeast cells (at 10^8^ cells/mL). The authors used bubbling water columns combined with SPE-GC-MS to capture gasses emitted during the fermentation and to analyze them. In order to determine the yeast capability to bind volatiles, yeast cells after contact with the volatiles in question were washed with solvents, and the solvents were analyzed using SPE-GC-MS. This suggests that late dry-hopping (when there are not as many suspended yeast cells, as during the primary fermentation) may be beneficial in releasing hydrocarbon-derived aromas in beer. Haslbeck et al. [50] found that rising temperatures in the tested ranges of 1 to 20 °C generally resulted in the increased extraction of hydrocarbons (but to different degrees for individual compounds). These results on the temperature-dependent extraction of hydrocarbons were confirmed by von Terzi [91]. Both authors used GC-FID for the hop essential oils analysis and HS-GC-MS for beer volatiles. In a study by Salamon et al. [53], the maximum concentration of β-myrcene (**1**) was reached after 34 h (301 μg/L) of dry-hopping, and its concentration in the beer stabilized after 44 h (215 μg/L) of the process. The levels of β-myrcene (**1**) were therefore above the sensory threshold (Table 2). The authors used 30 s of stirring to suspend the hop particles during dosing and the HS-GC-FID analysis.

Wolfe [95] reported that the duration of the dry-hopping process did not appear to have a significant effect on the overall final beer aroma, as evaluated with the sensory analysis. When static dry-hopping was investigated, the author found no statistically significant differences between 6 h and 12 days of the process duration. The temperature of the process probably has a small impact on the extraction of terpene alcohols, as confirmed by the results of Mitter and Cocuzza [93] (GC—own method). In the authors’ study, the concentrations of linalool in dry-hopped beer at 4 °C and 20 °C were at the same level on the 4th day of the process. The results of von Terzi [91] confirmed that the effect of temperature is small. A minimal effect of temperature (1 °C, 4 °C, and 20 °C) on the extraction of polar compounds was also found by Haslbeck et al. [50]. The ketones (2-nonanone (**10**) and 2-undecanone (**11**)) evaluated by Schnaitter et al. [56] underwent a significant degree of extraction already on the first day of the process (SPME-GC-MS). In the case of polyfunctional thiols, Reglitz et al. [96] (GC-GC-TOFMS) reported that most of the extraction of 4MMP occurred between the first and second day of dry-hopping, while the increases up to the eighth day of hopping were described as minimal. The temperature dependence of the extraction rates of the other compounds has not been investigated so far. However, since these compounds tend to be hydrophilic, their extraction rate should be much higher than that of hydrocarbons, and it is likely that they will readily pass into the beer [11,50,53,55,56,91,93,94,95].

### 4.3. Early vs. Late Dry-Hopping

Early dry-hopping can result in a significant loss of volatile compounds, especially hydrocarbons, due to ongoing fermentation processes. These compounds may evaporate from the beer with volatilized CO_2_ or may be adsorbed onto yeast cells that are largely suspended in the beer during fermentation, as is the case with β-myrcene (**1**). Takoi et al. [46] also reported a positive effect of later dry-hopping (day 3 of fermentation vs. day 6 of fermentation). Later hopped beers were characterized by higher levels of geraniol (**4**); isobutyric esters (isoamyl butyrate (**28**), isobutyl butyrate, and 2-methylbutyl butyrate); and ethyl heptanoate (**20**). The terpene alcohols assessed (linalool (**3**), β-citronellol (**6**), and nerol (**5**)) showed no significant changes. The positive effect on the concentrations of volatile compounds was also corroborated by Haslbeck et al. [79]. According to von Terzi [91], due to the high volatility of hydrocarbons, late dry-hopping can enhance the sensory characteristics associated with these compounds, such as herbal and balsamic. In contrast, an early addition can result in more pronounced floral, fruity, and citrus aromas [46,55,79,91,97].

### 4.4. Chemical Composition of Wort

In a study by Cocuzza et al. [24], the authors studied the variations in the extraction rates of compounds as a function of the alcohol concentration in beer. The range tested was 0.5% to 10.5%. The authors used a semi-dynamic hopping process (stirring the vessels twice) at 5 °C for 2 weeks. The doses used were 217–242 g/hL. Varying the alcohol content had no significant effect on the changes in the total polyphenol content, humulinones, iso-α-acids, and terpene alcohols contents. The latter were already highly extracted (77–91%) at an alcohol concentration of 0.5%. Increasing the alcohol concentration improved the extraction of α- and β-acids, xanthohumol, monoterpenes, sesquiterpenes, and ketones. For each 1% of alcohol, the content of α-acids increased by 5 mg/L; for β-acids, it was less than 1 mg/L. The alcohol content had no influence on the pH changes caused by the hopping process. In the case of the tested esters (geranyl acetate (**13**), isobutyl isobutyrate (**23**), 3-methylbutyl isobutyrate (**22**), and 2-methylbutyl isobutyrate (**24**)), they showed high extraction rates even at 0.5% alcohol, which increased up to 3.5% alcohol. No significant changes were observed above this level. Increased extraction of the hydrocarbon β-myrcene (**1**) with an increasing alcohol concentration was also observed by Haslbeck et al. [50] and von Terzi [91].

### 4.5. Other Factors

Mitter and Cocuzza [93] pointed out that a key factor affecting the extraction efficiency of hop-derived compounds is how the hops are added to a vessel. Sometimes, hops are added in various types of bags to facilitate their subsequent separation from the beer. In the study, the use of loose pellets resulted in nearly 50% higher levels of extracted linalool compared to hops in a bag [93]. For α-acids, the difference was more than double. The extraction rate of linalool was also influenced by the hop form; when hop cones were used, the concentration of this compound in the finished beer was 50% lower compared to pellets. The significant influence of the hop form was also reported by Wolfe [95], where beers hopped with pellets were characterized by a faster extraction of hop-derived compounds compared to hop cones. The levels of hop-derived compounds in the beers were also higher when pellets were used. Additionally, comparing static and dynamic (stirred) dry-hopping, the authors noted that the latter achieved a higher extraction rate of volatile compounds compared to the static method (even with a long hopping time). The aroma of beers subjected to dynamic dry-hopping was found to be significantly more intense after 6 h of the process, even compared with more than 12 days by the static process. Even 4 h of agitated dry-hopping was sufficient to fully extract the hydrocarbons from the hops. A potential disadvantage of agitated samples is a higher bitterness and astringency, which the authors attributed to the higher extraction of polyphenols. The authors noted, however, that the separation of suspended hops from such processes can be difficult when dynamic hopping is used. According to Vollmer et al. [98], the use of hops stored under aerobic conditions can intensify the perception of attributes such as ‘woody’ and ‘herbal’ in dry-hopped lagers, as assessed in the sensory evaluation. However, such hops can significantly increase perceived bitterness. According to the authors, this is due to the higher content of oxidized hop resins, namely humulinones and hulupones (Figure 1). An increase in the humulinones content with an increasing HSI (Hop Storage Index) was confirmed by Rutnik et al. [52]. The HSI measures the freshness of hops using a dimensionless index calculated from the absorbance of the alkaline methanolic extract of hops at 275 and 325 nm. At 325 nm, primarily α- and β-acids are detected, whereas, at 275 nm, decomposition and oxidation products are identified. Fresh green hops have a HSI ranging from 0.2 to 0.25, and the value of this parameter increases with the time as the hops age. It is worth noting that there may be differences in the initial HSI among different hop types. The HSI increase in oxidized fractions occurs up to a certain point; after which, the content of these compounds decreases, and the bitterness quality declines with further increases in the HSI. In beers dry-hopped with aged hops, the aroma quality declines, which is associated with the deceasing content of volatile elements in hops essential oil. The authors suggested that the maximum HSI for hops intended for dry-hopping should be less than 0.5. The authors also found that hops with a HSI above 0.6 can cause gushing [52,93,95,98].

An important phenomenon when considering beer aroma is the synergistic effects that are known to occur between the compounds. These effects can significantly enhance the perception of a given aroma or can alter the sensory characteristics of a given mixture of compounds. These effects are known to occur within terpene alcohols. Other groups of compounds can also significantly alter the perceived character of an aroma. Takoi [99] suggested that branched-chain fatty acids (isobutyric acid, isovaleric acid, and 2-methylbutyric acid) can enhance the aroma of monoterpene alcohols and that even threshold amounts of fatty acids are sufficient to observe this effect. The author suggested that the interaction between these compounds may be responsible for the tropical aroma found in some dry-hopped beers. Fatty acids can originate from yeast metabolic processes, as well as from the raw materials used in beer production (including hops). In another study by Takoi [100], the addition of isobutyric acid and 2-methylbutyric acid to a mixture of terpene alcohols increased the ‘tropical’ sensation during the sensory assessment. The addition of isovaleric acid increased the ‘fruity’ character. In another paper [61], the same author also reported a probable synergistic effect between 4-methyl-4-sulfanylpentan-2-one (at concentrations as low as 1.2 ng/L) and terpene alcohols (linalool (**3**) and geraniol (**4**)). These compounds, together with β-citronellol (**6**), may enhance the tropical aroma. According to Haslbeck et al. [50], the interactions between isobutyric esters and monoterpene alcohols can modify the citrus aroma in beer. Depending on the concentration of isobutyl isobutyrate (**23**), a synergism or antagonism with terpene alcohols was observed (altering the citrus aroma sensation). Additionally, volatile compounds can also alter the taste perception of beer [101]. Several authors have stated that hydrocarbons can affect the perceived bitterness and make it sharper [50,59,61,80,99,101,102].

## 5. Chemical Changes during the Process

Dry-hopping causes significant changes in the physical and chemical parameters of beer. The main parameter that changes is the pH of the beer, which increases. The finished beer should have a pH in the range of 4.0–4.5. It is known that a higher pH of beer intensifies the perception of bitterness. Maye et al. [92] stated that a pH increase of 0.1 pH units is comparable to an IBU increase of 2–3 units. The authors examined the increase in pH level during dry-hopping with Cascade hops using a hop dose of 0 to 2319 g/hL. The pH increase was nearly linear at 0.1 pH units for every 386.54 g/hL. Similar results were obtained by Lafontaine and Shellhammer [1] at an analogous dose of the same hop variety (~0.14 pH units) and by Bruner et al. [103] (0.1 pH units at 400 g/hL of Centennial hops). In contrast, a negligible pH increase was reported by Salamon et al. [53]. The results obtained by Kemp et al. [66] indicated that the pH increase continues during storage. It is not yet known which factor is responsible for this effect. It has been supposed that the extraction of various hop components may be responsible. Given the poor understanding of the mechanisms behind this phenomenon, further research is needed [1,29,53,66,92].

Another important change that can occur as a result of dry-hopping is the biotransformation of hop-derived compounds. Individual monoterpene alcohols can undergo biotransformation into other alcohols, e.g., linalool (**3**) into α-terpineol (**7**) or geraniol (**4**) into β-citronellol (**6**). This has been corroborated by the studies of many authors, such as King and Dickinson [104] and Takoi et al. [36]. Yeast can also synthesize various terpene alcohols, as demonstrated in wine yeast strains by Carrau et al. [105]. At present, no factors other than the yeast strain are known to affect the occurrence and intensity of the biosynthesis of these compounds. As mentioned, precursors of terpene alcohols and polyfunctional thiols are transferred during hopping and can be converted into their aroma-active forms by yeast activities [34,36,104,105].

Another change that may occur is a change in the qualitative composition of esters extracted from hops. Is it generally believed that methyl esters derived from hops are transesterified into ethyl esters in a beer environment. However, other important chemical changes have been postulated. Brendel et al. [106] postulated that the addition of hops induces the formation of ethyl esters of 2- and 3-methylbutanoic acid and methylpropanoic acid. The authors were unable to determine the exact cause of the formation of these compounds but postulated an enzymatic reaction. These reactions are probably catalyzed by native hop enzymes, as the formation of these esters was also observed when hops were added to water. The authors suggested that most of the esters are formed during the dry-hopping process by this phenomenon. The extraction of those already present in the hops is of secondary importance. The authors conducted analyses using HRGC/HRGC-MS for the quantification of esters in hop extracts and HS-SPME-HRGCxHRGC-TOF-MS for analyses of volatiles in the samples. The results obtained by Schnaitter et al. [56] (SPME-GC-MS) may be complementary to this information. The authors observed a significant increase in the concentrations of ethyl 2-methylpropanoate (**18**), ethyl 2-methylbutanoate (**16**), ethyl 3-methylbutanoate (**17**), and ethyl 4-methylpentanoate (**19**) in the final beer after the dry-hopping process. The authors explained this fact by the breakdown of bitter acids and the resulting esterification of 2-methylpropanoic, 2-methylbutanoic, 3-methylbutanoic, and 4-methylbutanoic acids with ethanol. Forster et al. [107] reported that, when hops with a high geranyl acetate (**13**) content are used in dry-hopping, small amounts of this compound are found in the resulting beers. These beers, however, contain high levels of geraniol, which may be released from its bound form [56,106,107].

In the case of polyfunctional thiols precursors, the results of some authors have indicated that a beer maturation process is required to release their volatile forms [108,109]. Chenot et al. [108] investigated the effect of several factors during fermentation that could influence the release of glycine and cysteine-conjugated thiols by *Saccharomyces cerevisiae*. Volatile sulfur compounds were not detected 7 days after the end of the fermentation process. The authors emphasized the importance of the maturation stage in the release of polyfunctional thiols from their precursors. In the study, the yeast strain was an important determinant of the degree of polyfunctional thiols release. The distinct strains were also characterized by different degrees of esterification of these compounds. At the same time, the S-cysteinated precursors were degraded to a greater extent than the glycine precursors (0.45 and 0.08% from Cys- and G-adducts, respectively). Worts with three different base extract levels were analyzed: 12, 15, and 17 °P (degrees Plato (°P) is used to quantify the concentration of an extract as a percentage by weight, e.g., a 10 °P wort contains 10 g of extract per 100 g of wort). A lower wort extract resulted in the release of a higher amount of sulfanyl alcohol (0.5–0.8% at 12 °P vs. 0.2–0.4% at 17 °P). Volatiles were extracted using PFT extraction with an Ag cartridge and were analyzed using GC-PFPD. *S. pastorianus* strains examined by Chenot et al. [110] were found to have a higher ability to release G-adducts (up to 0.35%). Lowering the fermentation temperature (12 °C) and the initial FAN (free amino nitrogen) content allowed the release of higher amounts of Cys-conjugated compounds. At the same time, the ester/alcohol ratio from cleaved Cys conjugates was increased [108,109,110].

## 6. Problems

### 6.1. Hop Creep

An important phenomenon that occurs as a result of dry-hopping is the resumption of the fermentation process, more commonly known as ‘hop creep’ or ‘the freshening power of hops’. Its occurrence was first reported in 1893. It is undesirable, because it results in a beer with a higher final alcohol content but a lower extract content (making the beer taste empty). It is particularly dangerous in non-pasteurized and unfiltered beers, as it increases the risk of overcarbonation. Overcarbonation affects the sensory characteristics of a beer and creates a risk of gushing. Highly affected beers may require the costly procedure of a product recall. In addition, in the worst cases, beer with this defect can pose a health risk to consumers, as too-high pressure in the bottle can cause it to explode. Beer with hop creep may also require longer maturation times as a result of the prolonged fermentation and possible production of vicinal diketones (diacetyl and pentanedione) [111]. It is known that the amylolytic enzymes present in hops are responsible for this phenomenon. These enzymes can break down the dextrins that make up the residual extract of beer. The dextrins are broken down into simpler sugars that can be further fermented by the yeast (if present in the beer). If hops are used during the ‘hot part of the production process’, these enzymes are most likely denatured. However, the degree to which refermentation occurs varies. One of the higher levels was detected by Kirkpatrick and Shellhammer [112], who added Cascade hops to a fully attenuated lager beer at 1000 g/hL at 20 °C. After 5 days, the decrease in extract content was 1 °P and finally reached 2 °P after 40 days. This resulted in an additional 1.3% (*v*/*v*) alcohol and 4.75% (*v*/*v*) CO_2_. The authors noted that the hop simple sugars content (Table 1) had no significant effect on the level of fermentable sugars in the beer. A similar negligible effect of simple sugars on the real extract was reported by Lafontaine and Shellhammer [1]. In their study, each 386 g/hL increased the real extract of the beer by ~0.07 % (*w*/*w*). Although Bruner et al. [113] did not find a correlation between hop creep intensity and hop variety, the results of other authors have presented a different point of view. Kirkpatrick and Shellhammer [112] investigated α-amylase, β-amylase, and amyloglucosidase, and limited the dextrinase activities of 30 hop varieties. All the varieties tested showed a significant variation in these activities, although the latter was found to be negligible among the varieties tested. Alpha-amylase activities ranged from 0.04 to 0.25 U/g (mainly 0.08 to 0.10 U/g), β-amylase from 0.14 to 0.21 U/g, and amyloglucosidase from 0.001 to 0.016 U/g. The different varieties therefore differed in their hop creep potential. This was also confirmed by Kirkendall et al. [114], who observed an increase in the glucose, fructose, maltose, and maltotriose levels during the 2-day dry-hopping process. The total increase in these fermentable sugars was 0.77 g/100 mL when fully fermented, which would result in an additional 0.39% (*v*/*v*) of alcohol and 1.9% (*v*/*v*) of CO_2_. In earlier studies, the authors noted that higher hop doses resulted in the release of higher amounts of fermentable sugars. This may be due to higher concentrations of amylolytic enzymes in the beer environment. This was partly confirmed by Werrie et al. [111]. They stated that this is only true when there are no live yeast cells in the beer environment. When yeasts are present, the loss of the process substrates (according to the authors, these are mainly maltotriose and maltopentanose) is the same for hop doses of 500 g/hL as for 2500 g/hL. The authors explained this fact by the retroinhibition of the enzymatic activities responsible for hop creep. The yeast present in the environment quickly metabolized the simple sugars released in the process, allowing the enzymatic reactions to continue.

Microorganisms and pests that may be present on hops are another potential factor that has been investigated in this regard. Cottrell [115] isolated microorganisms from hops that had the ability to degrade starch. Among the organisms isolated were *Klebsiella* sp., *Penicillium* sp., and *Alternaria* sp. They were isolated from the beers after 7 days, and they were found to be viable. However, the author concluded that they were not the direct cause of hop creep. The hop pathogens *Pseudoperonospora humuli* and *Podosphaera macularis*, commonly known as downy mildew and powdery mildew, are also suspected of enhancing the hop amylolytic activity. These microorganisms synthesize glycosyl hydrolases that can degrade complex sugars. To evaluate whether other plant additions could cause a similar phenomenon, Cottrell [115] used *Cannabis sativa* instead of hops in the dry-hopping process. Hops and *Cannabis* plants are closely related and belong to the same family (Cannabaceae). A similar overattenuation to hops was observed. This effect did not occur when the authors used oregano (*Origanum vulgare*) instead of hops. The authors suggested that both plants have similar genetic sequences encoding amylases responsible for this phenomenon. Rubottom et al. [116] reported that the hop creep potential of hops can be minimized by appropriate hop kilning temperature ranges. In the temperature range examined by the authors (49–82 °C), the decrease in amylolytic activity was found to be linear with the increasing temperature. Compared to a kilning temperature of 52 °C, the use of a temperature of 82 °C allowed the enzyme activity to decrease by ~2.6. However, the final effect of higher kilning temperatures depended on the initial level of enzymatic activity in the samples. In hops with high initial enzyme activity, hop creep occurred regardless of the kilning temperatures used. In hops with low initial enzyme activity, higher kilning temperatures may be an effective tool to reduce the potential for this phenomenon. Rubottom and Shellhammer [117] reported that higher kilning temperatures (54 vs. 65 °C) did not significantly affect the hop aroma and chemical composition, while it reduced the activity of enzymes associated with hop creep. Another tool that can help reduce hop creep is lower fermentation temperatures, resulting in lower enzyme activity. Yeast selection does not appear to have a significant effect on limiting this process. Bruner et al. [103] assessed 31 brewer’s yeast strains and found no significant differences in the occurrence of hop creep.

As for now, there is no known method to completely avert hop creep when dry-hopping is used. Determining the exact degree to which the beer is affected is also problematic without prior technological trials. One solution is to hop the beer after the yeast has been removed (e.g., using pasteurization or filtration). However, the inherent hop enzymes will degrade some of the beer dextrins to simpler sugars anyway. Based on current knowledge, such an approach will also prevent the biotransformation of hop compounds. Therefore, it seems most appropriate to design a beer by taking into account the possible refermentation processes due to dry-hopping [1,85,103,111,112,113,114,115,118,119].

### 6.2. Haze Formation

The colloidal stability of beer is an important factor affecting consumers’ perception of beer. Beer haze can be divided into two broad categories, i.e., biological and non-biological (colloidal) haze. The first is the result of microorganisms and can therefore be prevented through good manufacturing practices and processes such as pasteurization or filtration. The issue of colloidal haze is more complicated. The most common type of haze is chill haze (particle sizes between 0.1 and 1.0 μm), which forms at low temperatures (0 to 4 °C) but disappears when the beer is heated to ~20 °C. In contrast, permanent haze (particle sizes 1–10 μm) is also present in beer at higher temperatures. The small particles of chill haze can act as intermediate precursors for larger particles; therefore, chill haze can develop into permanent haze [120]. Chill haze is most likely formed by noncovalent bonds (e.g., hydrophobic interactions and hydrogen bonds), whereas, in the formation of permanent haze, covalent bonds are formed, which determine a greater haze persistence. Various compounds can cause this turbidity, e.g., dextrins, β-glucan, and arabinoxylan [121,122,123], but the most commonly analyzed and discussed are those that result from the interaction of malt-derived polyphenols and proteins [84,124,125,126,127]. Hydrophobic amino acids from malt are thought to be responsible for the formation of this type of turbidity, and proline and glutamine are mentioned the most often [128]. Phenolic compounds from hops and malt, such as proanthocyanidins, dimers, trimers, and tetramers of catechin, epicatechin, and gallocatechin, are likely to be involved in this process [121,129,130,131,132]. It should be noted that the profiles and concentrations of polyphenols depend on the type of malts [133]. One of the more important proposed mechanisms involves the reactions between polyphenolic compounds with binding sites in haze-active proteins that result in the formation of intermolecular bridges [120]. As the monomeric polyphenols in beer usually oxidize and polymerize [124], they form bigger particles that have more moieties that can bind to haze-active proteins. Approximately 70–80% of the polyphenols in beer are derived from malt, with the rest from hops [131,134]. However, in craft beers, where very high doses of hops are used, this ratio can shift. The issue of beer hazes has been discussed in more detail by many authors [29,119,127,135,136,137,138].

Huismann et al. [139,140] found in unfilterable dry-hopped India Pale Ale, that the haze was caused by β-glucans and proteins. The authors suggested that this type of haze was due to the presence of unfilterable yeast cell wall mannoproteins in the beer. In a study by Ferreira et al. [97], the severity of chill haze in dry-hopped beers stored for 24 months highly correlated with the content of flavan-3-ol monomers and oligomers in fresh beer and the degradation of these compounds during storage. A decrease of 1 mg/L in the content of these compounds resulted in an increase in the chill haze of 1.7 EBC unit (where EBC is a color unit of the European Brewery Convention, EBC unit = A430 × 25 × dilution factor) [97,139,140].

Haze is generally undesirable in the vast majority of beer styles, but one exception is the NEIPA style. In this style, high turbidity is often a desirable characteristic. To achieve high haze levels, a significant amount of high-protein malts are used in a grain bill, and, most importantly, significant amounts of hops are added several times during the cold phase of the process (e.g., at the early stages of fermentation). In a study by Maye and Smith [16], the turbidity of the NEIPA beers studied by the authors ranged from 119 to 1774 NTU (where NTU is the nephelometric turbidity unit, 1 NTU is a turbidity of 1 mg of formazine dissolved in 1 L of distilled water) (average 547 NTU), while the evaluated IPAs had a turbidity of less than 30 NTU. The turbidity composition of NEIPA beers was 35.7% protein, 11.1% carbohydrates, 3.4% polyphenols (tested as total polyphenols—ASBC method beer 35), and 0.9% saponified fatty acid. The beers were characterized by unprecedentedly high contents of components such as α-acids (17–72 mg/L, average 31 mg/L), xanthohumol (0.9–3.5 mg/L, average 2 mg/L), β-myrcene (**1**) (0.5–2.5 mg/L, average 1.4 mg/L), and β-acids (1–14 mg/L, average 5 mg/L). The authors stated that most of the NEIPA’s bitterness was caused by humulinones rather than iso-α-acids. It was suggested by the authors that the haze allowed the aforementioned compounds to dissolve in the beers at levels not typically found in this type of beverage. These results are interesting, because they show a different approach to haze in beers. Bolcato et al. [141] found that the haze formation is influenced by the yeast strain used and the time of the hop addition. It was found that the maximum polyphenol–protein haze formation occurred at pH ~4 [16,141,142].

### 6.3. Other Problems

#### 6.3.1. Grassy Flavor

It has been suggested that if dry-hopping is conducted for too long (more than a week), it may impart aromas described by Oliver and Colicchio [143] as vegetal, chlorophyll, and grassy. This can be facilitated by the high alcohol concentration. So far, this issue has not received much attention. It is mostly mentioned as a warning against too-long dry-hopping times. One theory is that, with long dry-hopping times, the plant matter in the beer begins to become extracted. According to Palmer [144], the use of improperly stored hops can also cause this effect. Compounds such as β-myrcene (**1**), geraniol (**4**), or linalool (**3**) can also impart vegetal or grassy aromas. Other potential factors may be the oxidation products of the fatty acids found in hops [145]. The enzymatic oxidation of unsaturated fatty acids, such as linoleic and linolenic acids, can result in the formation of hexanal and (E)-2-nonenal [146]. Another compound with an intensely ‘grassy’ aroma is (Z)-3-hexenal, found in fresh hops. (Z)-3-hexenal is formed from linoleic acid by an enzymatic reaction cascade [147,148]. The conditions occurring during long dry-hopping may allow the synthesis of this compound. According to a study by Hongsoongnern and Chambers [149], a large proportion of hexyl esters are characterized by a green-grassy/leafy aroma. However, the sensory thresholds of the compounds assessed by the authors were, in most cases, as high as 1000 mg/L. It is possible that synergistic interactions between compounds may lower the sensory threshold of the compounds responsible for this aroma. In addition, Saint-Eve et al. [150] showed that, in carbonated beverages, the release of CO_2_ increases the perception of aromas. A similar mechanism may occur in beers [11,144,149,150,151,152].

#### 6.3.2. Potential of Infection

Although hops are known to exhibit a wide range of antimicrobial properties, several types of microorganisms may be present on their surface. As mentioned, Cottrell [115] isolated organisms from the genera *Klebsiella*, *Penicillium*, and *Alternaria*. The author added them to beer with an alcohol content of 5.7% (*v*/*v*), and they showed viability after being stored in beer for one week. Kolek et al. [153] isolated *Pantoea agglomerans* from inflated hop bags. Van Vuuren et al. [154] intentionally infected the wort with this microorganism at concentrations of 10^3^ cells/mL and 10^6^ cells/mL at the beginning of fermentation. Notably, the second treatment increased the concentration of acetaldehyde, methyl acetate, diacetyl, 2,3-pentanedione, and dimethyl sulfide in the finished beer (the authors used headspace GC-ECD/FPD). However, the contamination of beer with hop-derived microflora is rather difficult to achieve. The treatments that hops undergo after harvesting reduce the number of microorganisms on the hop surface. Beer after the primary fermentation is a highly unsuitable environment for most microorganisms due to its low pH (3.5–4.5), significant alcohol content, trace amounts of simple sugars, and, if correctly managed, low oxygen content. Even when hops are added before yeast, the amounts of yeast added are high enough that contamination from this source should not pose a significant risk. It would therefore appear that the application of good manufacturing practices is sufficient to avoid this problem [29,115].

#### 6.3.3. Flavor Stability

A significant problem with dry-hopped beers is their low aroma stability. The aroma and sensory characteristics of beers subjected to this process undergo significant changes during storage. This can be particularly noticeable in beers with high doses of hops [16]. All of the bitter compounds discussed are lost to some extent during storage. There is a significant decrease in the content of humulinones. This was confirmed by a study in which the authors studied the changes that occurred when beers were stored at different temperatures (0, 3, and 20 °C) for different periods of time (3, 6, and 10 months), using HPLC-UV to determine the contents of the analyzed substances [66]. The most significant reductions in humulinones and α-acids were observed during the first 3 months of storage at 20 °C. However, over the subsequent months, the degradation of these compounds was minimal. Interestingly, after 10 months of storage at 20 °C, the levels of humulinones and α-acids actually increased compared to the previous period. When the beers were stored at 3 °C, the decline in the contents of these compounds was minimal. Ferreira and Collin [155] analyzed the methanolic extract of beers using RP-HPLC-UV and found even more significant reductions in the contents of selected compounds when beers were stored at 20 °C for a period of two years. The average losses of humulones and humulinones were 91% and 73% of their original contents, respectively. In the case of the total polyphenol content, Mikyška et al. [156] reported that the main decrease occurred during the first 2–3 weeks of maturation in the 6 weeks studied (analysis conducted according to Analytica EBC methods). However, it should be noted that these were not dry-hopped beers. There are no studies on this topic for dry-hopped beers. Regarding the volatile compounds, Kemp et al. [66] also investigated their fate under the above-mentioned conditions using SPME-GC-MS. They reported a significant decrease in the esters content (2-methylbutyl isobutyrate (**24**), 2-methylbutyl 2-methylbutyrate (**25**), methyl geranate (**26**), 2-methylbutyl propanoate (**27**), and 3-methylbutyl butyrate (**28**)) during storage. The exception was 2-methylbutyl 2-methylbutyrate (**25**), which content increased and was the highest after 6 months of storage. The content of 2-nonanone (**10**) increased, probably due to the oxidation of 2-nonanol. The contents of hydrocarbons such as β-myrcene (**1**), humulene (**29**), caryophyllene (**30**), limonene, perillene, α-selinene, and γ-muurolene decreased during storage. For the first three compounds listed, the losses were as high as 80% after the first 3 months. The contents of the remaining compounds decreased more gradually until the 10th month of the study. After 3 months of storage, the linalool content was the lowest, but after 6 and 10 months, it was similar to that of fresh beers, probably due to the oxidation of the other beer components. The content of geraniol remained stable throughout the storage period but dropped significantly after 10 months. In general, lower temperatures provided a higher stability of hop-derived volatile compounds. The authors observed such a relationship for esters and hydrocarbons (for example, the decrease in the β-myrcene (**1**) concentration at 20 °C was 87%, while, at 3 °C, it was 60%). The effect of the storage temperature was less profound on linalool (**3**) and geraniol (**4**). Interestingly, storage at 3 °C resulted in higher concentrations of these terpene alcohols than in fresh beers. Similar results were obtained by Guan et al. [157]. They used forced aging that corresponded to 12 months of storage and SPME-GC-MS analysis for volatile compounds. The aged beers scored significantly lower in the sensory evaluation. The authors also found an increasing concentration of linalool, while the concentration of geraniol decreased slightly. In the case of hydrocarbons, the content of β-myrcene (**1**) decreased by as much as 98.5%, while the other terpenes (caryophyllene (**30**), humulene (**29**), and β-pinene (**2**)) decreased by an average of 50%. The loss of the evaluated esters was as high as 96.6%, except for isoamyl acetate, which content increased 28.81 times. The content of 2-undecanone (**11**) decreased by about 60%. The significant loss of 2-methylbutyl isobutyrate (**24**) was confirmed by the study of Rettberg et al. [63] (SPME-GC-MS/MS). After 24 weeks at 20 °C, the concentration of this compound was reduced by 80%, while, at 4 °C, the loss was still 60%. Similar results were obtained by Drexler et al. [158]. In their studies, after 90 days of storage, the aromas of beers stored at 4 °C were rated significantly better during the conducted sensory analysis than those of beers stored at 28 °C.

Beer aging does not seem to have a significant effect on the concentration of humulene epoxides [159] (HS-SPME). Minimizing oxygen access to beer during its production and storage is an important issue in beer production. During dry-hopping, it is possible to introduce some oxygen. However, according to the results of Barnette et al. [160], although the addition of oxygen adversely affects the aroma of the beer by reducing the intensity of tropical and citrus aromas (sensory analysis), it does not significantly reduce the concentration of hop-derived monoterpenes (SPME-GC-MS). An additional loss of volatiles can occur due to the adsorption to the cap liners when beers are stored in bottles. This was reported by Kemp et al. [66], where the authors found esters and β-myrcene (**1**) to be susceptible to such adsorption [63,66,155,157,158,159,160,161].

The release of polyfunctional thiols during maturation and aging is still poorly understood. Nizet et al. [162] reported that compounds belonging to this group were detected in bottle-fermented beers (GC-MS and GC-PFPD). Chenot et al. [108] studied the changes in cysteinated and glutathionylated sulfanylalkyl alcohols during the beer production process. Free forms of these compounds were detected only after maturation (GC-PFPD). According to Tran et al. [163], the release of cysteinated thiols can occur during the first months of storage, even in the absence of yeast cells. However, after one year of storage, the concentrations of sulfanyl-4-methylpentan-2-one, 3-sulfanylhexan-1-ol (3SHol), and 3-sulfanyl-3-methylbutan-1-ol (3S3MBol) were significantly reduced (assessed by GC-PFPD and GC-MS). Unfortunately, the release of the hop-derived polyfunctional thiols from their precursors and changes in their contents is still a new and understudied issue [108,162,163].

Overall, the current knowledge about the stability of the flavor and aroma of dry-hopped beers suggests that they should be stored at lower temperatures (4 °C) to preserve their sensory characteristics.

## 7. Conclusions

During dry-hopping, the individual hop flavor components are extracted to varying degrees. The final aroma of the beer after the process is influenced by numerous factors, including the hop variety, its form and parameters, hop dosage, temperature, and time of the process, as well as the alcohol content of the beer. The current knowledge indicates that, in addition to hydrocarbons, other aroma compounds are extracted from hops to a high degree. The literature also confirmed that this happens quickly, and very long dry-hopping times do not seem to make much sense. In the case of dry-hopping, it is important to note two undesirable phenomena: hop creep and pH rise. While pH changes can be corrected, there is currently no known way to prevent hop creep. The significant loss of certain compounds from dry-hopping is also an important issue. The current literature suggests that dry-hopped beers should be stored at low temperatures (4 °C) to avoid significant changes in the flavor and aroma.

## Data Availability

Not applicable.

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
