# Peer review of "Effects of Dry-Hopping on Beer Chemistry and Sensory Properties—A Review"

_molecules, 2023, doi:10.3390/molecules28186648_

Round 1

Reviewer 1 Report

Some general remarks:

Please correct and unify the concentration units used: %, ppm, lb/bbl.

Please try to describe the analytical methods used by the authors you cite to understand better the concentrations/quantitation data of different molecules listed in your paper (Table 2 and elsewhere): HPLC?, GC?, ... ? Try to interpret these data from the view of different methods if used.

Please try to use a more profound description of the polyphenols data you mention (type of polyphenols, methods used (spectroscopic (total polyphenols) or HPLC-UV, MS/MS, ...).

Please try to briefly describe the procedure of adding the dry-hops. You only mention the three methods without telling them: static, dynamic, and stirred.

Please try to use a more descriptive and targeted title: Effects of dry-hopping on ….. – a review.

Reviewer 2 Report

 The review covers the broad range of chemical compounds and  chemical interactions occurring in the hop and during the process of  dry-hopping, as a method applied during the beer production in order to preserve the aroma of the final product.  In addition, it addresses the potential problems that may arise when dry-hopping is employed, focusing also on the important chemical aspects. The review is quite detailed, comprehensive, scientifically based and easy to follow, written in excellent English. Minor revisions are needed and suggested throughout the manuscript in the form of comments.
